# Polarizability-Dependent Sorting of Microparticles Using Continuous-Flow Dielectrophoretic Chromatography with a Frequency Modulation Method

**DOI:** 10.3390/mi11010038

**Published:** 2019-12-28

**Authors:** Jasper Giesler, Georg R. Pesch, Laura Weirauch, Marc-Peter Schmidt, Jorg Thöming, Michael Baune

**Affiliations:** 1Chemical Process Engineering, Faculty of Production Engineering, University of Bremen, Leobener Straße 6, 28359 Bremen, Germany; j.giesler@uni-bremen.de (J.G.); lweirauch@uni-bremen.de (L.W.); thoeming@uni-bremen.de (J.T.); mbaune@uni-bremen.de (M.B.); 2MAPEX Center for Materials and Processes, University of Bremen, 28359 Bremen, Germany; 3Department of Engineering, Brandenburg University of Applied Sciences, Magdeburger Straße 50, 14770 Brandenburg an der Havel, Germany; schmmarc@th-brandenburg.de

**Keywords:** dielectrophoresis (DEP), microparticles, polystyrene, chromatography, interdigitated electrodes, microfluidic, separation

## Abstract

The separation of microparticles with respect to different properties such as size and material is a research field of great interest. Dielectrophoresis, a phenomenon that is capable of addressing multiple particle properties at once, can be used to perform a chromatographic separation. However, the selectivity of current dielectrophoretic particle chromatography (DPC) techniques is limited. Here, we show a new approach for DPC based on differences in the dielectrophoretic mobilities and the crossover frequencies of polystyrene particles. Both differences are addressed by modulating the frequency of the electric field to generate positive and negative dielectrophoretic movement to achieve multiple trap-and-release cycles of the particles. A chromatographic separation of different particle sizes revealed the voltage dependency of this method. Additionally, we showed the frequency bandwidth influence on separation using one example. The DPC method developed was tested with model particles, but offers possibilities to separate a broad range of plastic and metal microparticles or cells and to overcome currently existing limitations in selectivity.

## 1. Introduction

Separating microparticles according to specific properties such as size, material, and shape is a research area of great interest for instance in cell or biomolecule manipulation [1,2,3,4,5] and waste recovery [6,7]. To separate microparticles, field-flow fractionation [8], gel electrophoresis [9], and size-exclusion chromatography [10] are state-of-the-art approaches. A major drawback of these approaches is their low throughput or low selectivity for particle mixtures with similar separation properties (e.g., shape, density) below a particle size of 10 μm [11,12,13]. Dielectrophoresis (DEP), which is referred to as the movement of polarizable particles in an inhomogeneous electric field, offers an alternative tool to address a wide range of particles and at the same time is able to achieve relevant throughputs [14,15]. The dielectrophoretic force not only depends on one specific property of a particle, but on a variety of particle properties, such as size [16,17], permittivity, and electrical conductivity [1], allowing for multi-dimensional particle fractionation. Apart from established DEP concepts such as field-flow fractionation [17,18], filtration [19], selective trapping (e.g., insulator-based dielectrophoresis) [20], dielectrophoretic particle chromatography (DPC) is a promising concept to achieve high throughput separation of particles. Since DPC was introduced by Washizu et al. [5], different approaches were done using selective trapping of particles [21,22], packed bed columns [23], or stepwise change of the frequency [24]. DEP chromatography proved to be very successful in isolating circulating breast tumor cells (CTCs) from blood [25] at a very low concentration. Such studies later led to the development of a clinical high throughput device to separate CTCs from blood samples [26,27]. Aldaeus et al. [28] developed an analytical model for a DPC device that was based on multiple trap and release cycles for fractionation. A related technique to manipulate micrometer sized particles is using traveling wave dielectrophoretic separators [29,30]. In these microfluidic devices, a 90${}^{\circ }$ phase angle is present between adjacent electrodes, which changes the dielectrophoretic movement a particle experiences [31,32]. Such traveling wave systems offer versatile particle separation techniques, but are usually complex to fabricate and operate [30,33]. The other presented dielectrophoretic chromatography techniques have in common that they depend on strongly diverging polarizabilities (e.g., one type of particle showing positive dielectrophoresis, whereas the other particles show negative dielectrophoresis or exhibit no dielectrophoretic movement). This requirement limits the applicability when addressing particle mixtures with less pronounced differences in polarizability. Addressing binary (or more) mixtures in which there is heterogeneity in the two (or more) classes is even more complex, especially when the cross-over frequencies of the classes are so close that the heterogeneity causes an overlap (an example is the separation of cells according to only small differences in their expression).

Here, we introduce the novel concept of frequency modulated dielectrophoretic particle chromatography. The frequency of the applied field changes constantly to exploit small differences in the dielectrophoretic mobilities of target particles. In this technique, by switching the frequency, we switch between positive and negative dielectrophoretic movement of target particles to generate multiple trap-and-release cycles, which leads to a polarizability dependent chromatographic separation. In principle, this allows separating particles that even show only minute differences in their polarizability and to separate mixtures with heterogeneity in the classes. The simplicity of our approach allows for a simple fabrication and operation and could be easily scaled up by using different ways to introduce the electric field gradient (for example using a porous medium as demonstrated in our recent work [14]).

## 2. Method

### 2.1. Theory

In classic chromatographic processes (e.g., gas chromatography), mixtures are separated due to different interactions of the sample and stationary phase, leading to characteristic retention times for each class in the sample. In dielectrophoretic particle chromatography, the stationary phase is represented by the inhomogeneous electric field rising over interdigitated electrodes. The electrode chip forms the bottom of a microfluidic device, where a polydimethylsiloxane (PDMS) channel is used as the separation column. The microparticle suspension is injected into the flow chamber and further transported by a carrier flow. The electrodes are connected to an AC voltage source to generate a highly inhomogeneous electric field. This gives rise to a dielectrophoretic force on the particle caused by the action of the inhomogeneous field on the induced dipole (or multipole) of the particle. In the simple point-dipole approximation, the dielectrophoretic force $F_{\mathrm{DEP}}$ can be expressed as:(1)FDEP=πrp3εmReε˜p−ε˜mε˜p+2ε˜m∇|E|2,
with $r_{p}$ representing the particles radius, ${\nabla |E|}^{2}$ the electric field gradient squared, and ${\tilde{\varepsilon }}_{p}$ the complex permittivity of the particles and the medium (${\tilde{\varepsilon }}_{m}$), respectively. The velocity due to dielectrophoresis, $v_{\mathrm{DEP}}$, in a stationary fluid can be calculated by dividing the dielectrophoretic force by the friction factor $f^{*}$:(2)vDEP=μDEP∇|E|2=πrp3εmReε˜p−ε˜mε˜p+2ε˜m∇|E|2f*.

Here, $\mu_{\mathrm{DEP}}$ is the dielectrophoretic mobility, which not only provides the direction of the movement of the microparticles, but incorporates the radius of the particles and fluid properties additionally. The direction of the DEP force can be determined by calculating the real part of the Clausius–Mossotti factor Re(CM):(3)$$\mathrm{CM}=\frac{{\tilde{\varepsilon }}_{p}-{\tilde{\varepsilon }}_{m}}{{\tilde{\varepsilon }}_{p}+2{\tilde{\varepsilon }}_{m}}.$$

The complex permittivity expands the permittivity ε of a material and incorporates the material’s conductivity σ and the angular frequency ω of the electric field:(4)$$\tilde{\varepsilon }=\varepsilon_{0}\varepsilon_{r}-i\frac{\sigma }{\omega }.$$

For low frequencies, Re(*CM*) is dominated by the conductivity of the material. With increasing frequency, the permittivity becomes more important. When particles are less polarizable than the surrounding medium (Re(CM)<0, negative DEP), they move against the electric field gradient and towards low field regions. On the contrary, more polarizable particles (Re(CM)>0, positive DEP) are directed with the gradient towards field maxima. In the current setup, field maxima are located close to the edges of the interdigitated electrodes at the bottom, and local field minima can be found at the top of the channel. Depending on the polarization, particles are either attracted to the edges of the electrode (positive, pDEP) or to the top (negative, nDEP). Therefore, the movement direction will be strongly affected by the applied field’s frequency due to the frequency dependence of Re(*CM*) (Equation (Equation 4)). Particles can become trapped in potential wells (field extrema) due to DEP and can adhere to the walls of the device when they reach them.

The conductivity of small insulating particles (such as the polystyrene particles that are used in this study as a model) is dominated by their surface conductance $K_{S}$ [34]:(5)$$\sigma_{p}=\frac{2K_{S}}{r_{p}}.$$

Usually, $K_{S}$ is assumed to be around 1 nS for polystyrene particles [35]. Equation (Equation 5) leads to a (with increasing particle diameter decreasing) net conductivity of polystyrene particles ($1\;\mu m<d_{P}<10\;\mu m$) of around 4μS/cm–40μS/cm, which is higher than some low conductive DEP buffers. This allows for positive DEP manipulation at low frequencies of even electrically insulating particles, when they are smaller than a certain threshold diameter.

To evaluate the resolution of a chromatographic separation, $R_{S}$ can be calculated [28],
(6)$$R_{s}=\frac{\Delta t}{\frac{1}{2}\left(w_{1}+w_{2}\right)},$$
with Δt as the separation time between the maximum values ($I_{\max}$) of two peaks and $w_{x}$, the width of the two residence time distributions. The width is defined as the distance in time between the half maximum values (FWHM).

### 2.2. Device Operation

The device proposed here, a microfluidic channel with interdigitated electrodes at the bottom of the channel (Figure 1b), uses periodic changes from pDEP to nDEP or vice versa to separate particles with respect to their polarizability. Since the polarizability of a particle directly depends on the frequency of the electric field, constant frequency changes (Equation (Equation 7)) can be used to manipulate the particles’ position in the separator. To achieve a retardation, due to either nDEP or pDEP, particles are dragged out of the fast streamlines in the center of the channel to streamlines with low fluid velocity at the bottom or top. Then, when the frequency changes, the pDEP or nDEP effect is reversed, and particles are pushed back into the faster streamlines in the center of the channel. Depending on the strength of the interaction of a particle with the field (i.e., the absolute value of Equation (Equation 3)), particles with different polarizabilities experience different retardation. Unlike DEP field-flow fractionation, there is no particle equilibrium position. Here, the periodic change of frequency leads to a constant change of the particles position and, therefore, depending on the polarizability of a particle, to a different average velocity.

In case a particle gets trapped in potential wells because of dielectrophoresis or adheres to the surface of the channel due to non-specific adsorption, a particle resuspension requires a force pointing away from the wall, which is in our case again DEP. Naturally, to reverse the trapping movement, particles trapped by pDEP now have to experience nDEP and vice versa (Figure 1a). Especially for particles trapped at the bottom of the channel, a resuspension via an external force becomes important, since no gravitational force contributes to their remobilization. Further, as the particles’ diameter decreases, the gravitation force becomes less important and therefore may not be sufficient to resuspend small particles close to the ceiling of the channel. To achieve a retardation of the particles and consequently a chromatographic separation, it is in general not necessary to fixate particles at the bottom or ceiling. To generate an increase in retention time, particles are just required to be transported into regions of low fluid velocity, which are present at the bottom (transport via pDEP) or the ceiling (nDEP) of the channel. Apart from the approach taken here, which is to modulate the frequency to reverse the particle polarization and the DEP force vector’s direction, in principle, it would also be possible to change the polarization by changing the medium’s conductivity.

Depending on the particle’s Clausius–Mossotti factor as a function of frequency (Equation (Equation 3)), three different scenarios can be distinguished (Figure 1a): (I) A particle shows substantial more pDEP than nDEP during the modulation spectrum and therefore predominantly moves towards the bottom of the channel, where the electrodes induce a high electric field strength. Since the fluids’ velocity close to the bottom is low, particles are slowed down by the lower fluid velocity or by getting reversibly trapped at the electric field maxima. The particles are then pushed away from the electrodes by nDEP when the frequency changes. This scenario effectively increases the particle’s residence time. (II) When a particle exhibits a balanced pDEP and nDEP movement, the retardation is less pronounced. These microparticles travel towards high field regions when the CM factor is positive and away from them when it is negative. Due to their constant movement orthogonal to the fluid-flow direction, they spend less time in regions with low fluid velocities and therefore are eluted fast. (III) If nDEP outweighs pDEP, particles are predominantly pushed towards low field regions, which here are present at the channel’s ceiling. Like in Scenario I, only low fluid-flow is present at the field minima, and the particle’s residence time is going to be enlarged. Although the polarizability of particles from Scenarios I and III is different, retention times can be the same. Nevertheless, since the extent of retardation depends on the chosen process parameters (e.g., frequency, voltage), a separation can be possible with a different set of parameters (Figure 1e,f).

Here, the frequency of the applied sinusoidal voltage was modulated using a triangle-shaped function. This allows changing the frequency of the electric field constantly between two values in a controllable time. Consequently, the frequency *f* can be described as a function of time *t*:(7)$$f\left(t\right)=f_{A}\mathrm{tri}\left(2tf_{\mathrm{mod}}\right)+f_{0}$$
with $f_{A}$ as the amplitude of frequency modification, tri(x) as the triangle function, $f_{\mathrm{mod}}$ representing the modulation frequency, and $f_{0}$ for the offset of the frequency modulation. As an example, for achieving frequencies between 30 and 270 kHz. the following set of parameters was used: $f_{A}=120\;\mathrm{kHz},f_{\mathrm{mod}}=300\;\mathrm{mHz}$, and $f_{0}=150\;\mathrm{kHz}$. Other modulation functions may also be suitable for achieving a separation.

In this study, we used polystyrene (PS) particles to demonstrate the functionality of the proposed technique. Due to their surface conductance (Equation (Equation 5)), PS particles show pDEP at low frequencies and nDEP at high frequencies. With the usually assumed $K_{S}=1\;nS$ [34,35,36] and a medium conductivity of $\sigma_{M}=1.2\;\mu S/cm$, the cross-over frequency from negative to positive DEP (Re(CM)=0 in Equation (Equation 3)) depends only on particle size (see Appendix A). The frequency dependent polarizability of the particles forms the fundamental aspect of this separation technique and can be used by varying the frequency over time periodically, as shown in Figure 2. These periodical changes from pDEP to nDEP generate multiple trapping and release cycles. The separation technique can also be used for other particle types that show frequency dependent polarizability.

Larger polystyrene particles showed pDEP in a smaller frequency bandwidth and, consequently, when varying the frequency as shown, for a shorter duration. Four different polystyrene particle sizes were chosen to demonstrate the separation effect. With our chosen frequency modulation from 30 kHz–270 kHz, 3 μm particles showed predominantly positive DEP, 6 μm particles a balanced pDEP/nDEP behavior, and 10 μm particles predominantly negative DEP. Further, we used 2 μm particles to assess the possibility to separate two particle types that both experienced predominantly pDEP.

### 2.3. Device Fabrication

The microfluidic device consisted of two main parts. The column was formed by a 2 mm wide meandering PDMS channel (height 80 μm, length 17 cm), which provided the walls and the top of the channel (Figure 1b). The bottom was formed by the electric field generating electrode chip. Both parts were bonded using an intermediate layer as described later. The PDMS channel were produced using an SU8 master mold (soft lithography). The interdigitated electrodes (electrode arm width and gap width 100 μm) were fabricated using standard cleanroom techniques. Full details of the fabrication method can be found in Appendix A.

The electrode covered glass slide was bonded to the PDMS channel using liquid PDMS (10:3, base:curing agent). PDMS was selected as the intermediate layer, because of its well known spinning curves, low toxic potential, and easy accessibility [37,38,39]. The PDMS mixture was spin coated at 6000 rpm for 330 s on the electrodes. Using these parameters, the thickness of the uncured PDMS layer should be below 3 μm [38]. Subsequently, the cleaned PDMS channel was manually aligned over the electrodes and placed onto them. The bonding was finalized by curing the intermediate layer at 80 ${}^{\circ }C$ for an additional two hours. The PDMS did not only allow bonding the electrodes to the channel, which proved to be unsuccessful in our lab using corona bonding; it also reduced the unspecific adhesion of the particles to the electrodes [40]. Since using PDMS as the intermediate layer creates a reversible bonding and PDMS channels are inexpensive to replace, several channels were used during the experiments, and no significant changes between them could be observed.

### 2.4. Experimental Setup

Two syringe pumps were connected to a manually actuated 4 way valve (H&S V-101D, IDEX Health & Science, LLC, Oak Harbor, WA, USA). One syringe pump (KDS-100-CE, KD Scientific Inc., Holliston, MA, USA) controlled the volume flow of the carrier fluid; the other pump (LEGATO 270, KD Scientific Inc., Holliston, MA, USA) provided the flow of the particle suspension (both 5 mL
h^−1^). In the normal position, the carrier flow was connected to the inlet of the separation column. To initiate the experiment, the valve was manually turned to allow a 2 s pulse of particle suspension to flow into the separator (Figure 1a). The injection in all experiments happened at t=10 s. The carrier fluid was pure water containing 0.02 vol% Tween20 (Sigma-Aldrich, Steinheim, Germany) to reduce particle–wall interactions, 0.003 vol% 0.01
mol
L^−1^ potassium hydroxide in deionized water to adjust pH, and potassium chloride to adjust the electrical conductivity to the desired value (1.2
μS
cm
^−1^). The particles were suspended in the same suspension as the carrier flow, but without adding potassium chloride.

Monodisperse fluorescent polystyrene particles (Fluoresbrite, Polysciences Europe GmbH, Hirschberg, Germany) of different sizes and colors (2 μm polychromatic red, 3 μm yellow-green, 6 μm polychromatic red, and 10 μm yellow-green plain particles) were mixed and diluted in the described solution.

The inlet of the channel was connected to the manually actuated 4 way valve via a capillary (inner diameter: 100 μm) with a length of about 17 mm. To allow a controlled injection of the particles (i.e., to avoid dispersion of the peak), the internal volume of the connection from valve to channel inlet should be kept as small as possible. The chosen (short and with small diameter) inlet capillary resulted in a volume of 135 nL, resulting in an average residence time of less than 100 ms in this capillary.

The electrodes were connected to a voltage amplifier (PZD2000A, TREK, Lockport, New York, NY, USA) controlled by a signal generator (Rigol DG4062, Rigol Technologies EU GmbH, Puchheim, Germany). The signal generator provided the functionality of frequency modulation inherently. The amplifier’s output signal was monitored using an oscilloscope (RIGOL DS2072A, Rigol Technologies EU GmbH, Puchheim, Germany). The amplification factor of the amplifier was not constant, but decreased with increasing frequency. The output decreased by 4.3% per 10 kHz, which resulted in exponential decay in the applied voltage. All stated voltages were measured at 30 kHz. This circumstance may be overcome by using a different amplifier in future experiments.

The different fluorescent stains of the particles allowed to easily distinguish between them. To observe the particles, an inverted microscope (ECLIPSE Ts2R-FL, Nikon Instruments Europe BV, Amsterdam, The Netherlands) was used. For observation, a 40,6-diamidino- 2-phenylindole/fluorescein isothiocyanate/tetramethylrhodamine isothiocyanate (DAPI/FITC/ TRITC, excitation: 387/478/555nm, emission: 433/517/613nm) triple bandpass was selected, which allowed observing at least three different types of particles at once. However, only two particle colors could be observed simultaneously, since the current optics inhibited the DAPI excitation. Videos of the fluorescence were recorded at the outlet of the channel using a color CMOS camera (GS3-U3-51S5C-C, FLIR Systems Inc., Wilsonville, OR, USA), which were further processed using MATLAB (see Appendix A, for further information). In MATLAB, the frames were segmented, resulting in different pictures for each particle and background. Finally, the intensity of each picture was counted and plotted over time.

## 3. Results and Discussion

Three different main experiments were conducted to demonstrate the different capabilities of the proposed separator: We firstly demonstrate the possibility to separate particles experiencing predominantly pDEP from particles with a balanced pDEP/nDEP behavior. This was done by separating 3 μm particles from 6 μm particles. We further show the separation of predominantly nDEP experiencing particles (10 μm) from particles experiencing balanced pDEP/nDEP (again, 6 μm particles). Finally, we show that even particles that both experience mostly pDEP in the modulated frequency spectrum can be separated by separating 3 μm particles from 2 μm particles. Appendix A provides Particle Image Velocimetry (PIV) data of 10 μm particles at 100 V_pp_ and 0 V to demonstrate the fluctuation of the velocity due to the nDEP effect. Further, Appendix A visualize the separation of 3 μm and 6 μm particles at 80 V_pp_ and 0 V. For such small particles, it was not possible to extract the velocity reliably from the video using PIV. Nevertheless, the velocity fluctuations due to the action of DEP were clearly visible for the 3 μm particles. Unfortunately, from the observation perspective and with the experimental methods at hand, it was not possible to infer if particles were slowed down because they were attracted to or pushed away from the electrode array.

For the 3 and 6 μm particles, without an electric field, both particles showed typical retention time distributions for a laminar flow without observing separation (as expected; Figure 3a). When applying a voltage with frequency modulation, we could observe a clear chromatographic separation for all investigated voltages, i.e., 60 V_pp_, 80 V_pp_, 100 V_pp_, and 120 V_pp_ (see Appendix A for the full dataset). To achieve separation, the frequency was varied between 30 kHz and 270 kHz in 3.33
s (full cycle length, 300 mHz). Various parameters for frequency modulation were tested in advance, but this set of parameters worked best. However, the influence of each parameter is not fully understood and needs to be investigated further. While we could observe separation at all voltages, the best resolution for the separation of 3 μm and 6 μm particles could be achieved at a voltage of 80 V_pp_, resulting in an average resolution of $R_{s}=3.60\pm 0.31$ (number of experiments N=4) (Figure 3b). To provide a visual impression of the separation of 3 μm and 6 μm at 0 V_pp_ and 80 V_pp_, see Appendix A.

At all investigated voltages (see Appendix A and Figure 3b), the 6 μm particles eluted earlier than the smaller particles, which showed a substantial delay with respect to measurements without the electric field. This was because the 3 μm particles showed predominately pDEP in the frequency modulation range and thus were substantially retarded due to the DEP interaction (Figure 2, blue line). Interestingly, the peak size of the 3 μm particles decreased significantly, which suggested that their retention time was dominated by DEP and not by their initial height in the channel, as was visible in the experiments without applied voltage. In contrast, the peak size and position of the 6 μm stayed almost the same, which was due to the balanced nDEP/pDEP ratio (Figure 2, orange line). This balanced nDEP/pDEP led to a negligible movement orthogonal to the fluid-flow direction over one cycle. Consequently, at low to moderate voltages, particles were only slightly retarded in the channel caused by moving along the different streamlines of the parabolic flow profile. Since the dielectrophoretic velocity increased with increasing electric field strength (Equation (Equation 2)) and therefore with increasing applied potential, we assumed that particles traveled greater distances orthogonal to the flow, eventually hitting either the electrode array or channel ceiling, as the voltage increased. We thus assumed that with increasing voltage, also 6 μm particles would experience retardation.

The calculations suggested (Figure 2) that the retardation of the 3 μm PS particles was based on their movement towards the interdigitated electrodes (pDEP). To investigate the effect of nDEP on the retention time, we separated 10 μm particles, which showed predominantly nDEP in the chosen frequency modulation spectrum (Figure 2), from the balanced 6 μm particles (Figure 4). This switch from pDEP dominated behavior, to an nDEP/pDEP -balanced behavior, to an nDEP dominated behavior with increasing particle size was due to the decreasing conductivity of polystyrene particles with increasing diameter (Equation (Equation 5)). We observed a chromatographic separation of 10 μm from 6 μm particles (see Appendix A for the full dataset) for 80 V_pp_, 100 V_pp_, and 120 V_pp_ at 30 kHz–270 kHz. As before, the 6 μm particles showed almost no change in their retention time, whereas the larger and less polarizable 10 μm particles showed substantial delay, which indicated a retardation due to nDEP. Appendix A shows PIV data of the 10 μm particles at 0 V and at 100 V_pp_ to demonstrate how their velocity periodically decreased and increased due to the nDEP effect. This periodic velocity fluctuation corresponded exactly to the applied frequency modulation.

Before we address the more challenging task of separating 2 μm and 3 μm particles that both experience pDEP in the modulation spectrum, we discuss the resolution for the separation of 6 μm from 3 μm and 6 μm from 10 μm (Figure 5). The resolution of the separation of 3 and 6 μm (Figure 5a, green) particles increased with voltage in all conducted experiments until a maximum at 80 V_pp_ was reached, after which the resolution decreased. This was because the retention time of bigger particles increased further with voltage (80 V_pp_: 27.92
s±1.74s to 160 V_pp_: 40.8
s±3.33s, both N=4), while the time of the maximum fluorescence intensity for the smaller particles was constant for all voltages investigated, as long as a voltage was applied. We suspect the increase of the retention time of the 6 μm particles was because of the increased covered distances orthogonal to the fluid-flow. As previously discussed, the higher field strength caused the particles to reach the walls or at least enter regions close to a wall with low fluid velocity (at the top and bottom of the channel) and to be retarded as a consequence.

This decrease in resolution was not observed for the separation of 6 μm and 10 μm particles (Figure 5a, turquoise). Although the retention times of the 6 μm increased monotonically with voltage, the resolution simultaneously increased with applied voltage. The even stronger increase in retention time of the 10 μm particles (80 V_pp_: 45.61
s±5.24 s to 120 V_pp_: 57.71
s±1.6 s, both N=4) compensated the increase from the 6 μm particles. We assumed that the 10 μm particles spent even more time in areas with low fluid velocity, and therefore, the retention time increased. Our PIV measurements (see Appendix A) indicated a periodic interaction of the particles with the electric field. However, the change in velocity was below 20%, which showed additional potential for increasing the retention time of the 10 μm particles.

To investigate the effect of the voltage on the resolution of the separation process further, particles with diameters of 2 μm and 3 μm were selected. As the mobility of both particles was close to each other, this posed a more ambitious separation problem. Using the same set of parameters as before, we could again observe a voltage dependence of the peak time (see Figure 5b and also Appendix A, for intensity profiles as a function of time). In contrast to the 3 vs. 6 μm and 6 vs. 10 μm experiments, the separation was low for all investigated voltages. This was because $\mu_{\mathrm{DEP}}$ for both particle types was low and very close to each other.

The separation of the 2 and 3 μm particles was improved, concerning peak width and peak distance, by changing the frequency between which was varied. Applying an offset of 50 kHz (now: 80 kHz to 320 kHz), the retention times of each particle type became more homogeneous (FWHM decreased) and the distance between the peaks increased (Figure 5b). Interestingly, the resolution was similarly low for both sets of frequencies for all voltages except for 120 V_pp_ and 160 V_pp_, but the retention times were significantly different. At 120 V_pp_, where the highest resolution using 30 kHz–270 kHz was achieved, the particles eluted almost 15 s later than when using 80 kHz–320 kHz at the same voltage (2 μm: 55.22
s±
6.94
s vs. 40.73
s±
0.75
s, 3 μm: 49.06
s±
5.32
s vs. 36.86
s±
0.84
s, both *N* = 4).

We propose that at the lower frequency (30 kHz to 270 kHz), both particles were dominated by pDEP (Figure 2) and thus showed a significant increase in retention time with increasing voltage. When the frequency set switched to 80 kHz–320 kHz, the pDEP/nDEP behavior was more balanced, i.e., both particles exhibited less pDEP and more nDEP in the modulation spectrum, causing them to interact less with the field and thus to elute earlier. As expected, the subtle differences in polarizability of 2 μm and 3 μm particles in this frequency bandwidth were more pronounced, resulting in a better resolution. Although the residence time of the particles was much shorter when applying 80 kHz- 320 kHz, the resolution stayed the same at 100 V_pp_ and increased even further to $R_{s}=1.25\pm 0.23,\left(N=4\right)$, when the voltage was set to 160 V_pp_, which did not occur using the lower frequency set. This highlights one of the potentials of our separation technique, i.e., the possibility to separate particles with very equal polarizabilities by tuning the frequency modulation according to the target particle’s polarizabilities. Again, the particles larger in diameter eluted earlier, and in contrast to the 6 μm particles, both particles showed an increase in retention time with respect to the measurements without the electric field and therefore without superimposed dielectrophoretic movement. No maximum in retention time was found for the 2 vs. 3 μm mixture at higher frequency, indicating that a further increase in voltage led to a further increase in resolution.

Since the dielectrophoretic velocity depended quadratically on the particle’s radius and the applied electric field, the size and voltage dependency was not surprising. Due to this, smaller particles accelerated less due to DEP. Additionally, the cross-over frequency at which the force switched from pDEP to nDEP was higher, i.e., small particles experienced pDEP for a longer duration per cycle. Consequently, small particles, once they came close to the interdigitated electrodes, remained there and thus in regions of low fluid velocity. The latter point should become more important as the residence time in the separation column increases (i.e., at a longer column length).

Both nDEP and pDEP can be utilized to induce a retardation of the suspended particles. As a consequence, particles with polarizability (e.g., one showing more pDEP, another one dominated by nDEP) can elute at the same time. However, as the frequencies and the modulation frequencies can be adjusted, the nDEP/pDEP ratio can be tuned, which should result in different retention times and lead to a chromatographic separation.

Despite the fact that the parameters were chosen by evaluating the mobility of the particles over the frequency and only model particles were evaluated, the technique can become a tool for chromatographic separation of arbitrary particles that show a frequency dependent polarizability. One major advantage of this technique is that the columns’ parameters were adjustable without actually changing the column itself. As shown, the electric field strength and the frequency bandwidth had an impact on the retention times and the peak width. Joule heating could disturb the separation when mediums with higher conductivities are used (e.g., cell buffer) [41]. To reduce the required voltage, by maintaining a similar electric field strength, the thickness of the isolating layer on the electrodes could be reduced. Promising alternatives to PDMS to achieve thinner coatings are polymers with a lower viscosity (e.g., SU-8). Additionally, since the electric field decreases with the height of the channel significantly, channels with a reduced height could be used.

## 4. Conclusions

We experimentally showed the separation of three binary mixtures of suspended particles using dielectrophoretic particle chromatography with a modulated electric field frequency. The current data further suggested that a separation of three different particle types (for example 2,3 and 6 μm) in a single experiment should be possible. Unfortunately, it was not possible to observe all three different kinds of particles at the same time with the current hardware. We believe that an increasing column length led to a better separation. In addition to this, when the injection valve was operated automatically (in contrast to the current manual operation), standard deviations should decrease significantly. The influence of other parameters such as the modulating frequency, the medium’s electrical conductivity, the linearity of amplification, and the carrier fluids’ volume flow are complex and not yet understood in detail. Comprehensive studies regarding their impact using experiments and simulations are under way. Nevertheless, we demonstrated the principle and discussed the effect of the applied voltage. We further showed how adapting the modulation frequency to the target particle’s polarizabilities further increased the resolution. In the proposed chromatography column, no single trap-and-release mechanism was used to achieve a chromatographic separation, but the particles showed different interactions with the permanently present and adjustable stationary phase. Although we only studied model particles in this study, the presented method allowed chromatographically separating arbitrary particles with frequency dependent polarizabilities. We believe that the presented technique can potentially separate particle mixtures that are traditionally difficult to separate, for instance cell separation in liquid biopsy or the recovery of precious materials from waste streams.

## Figures and Tables

**Figure 1 micromachines-11-00038-f001:**
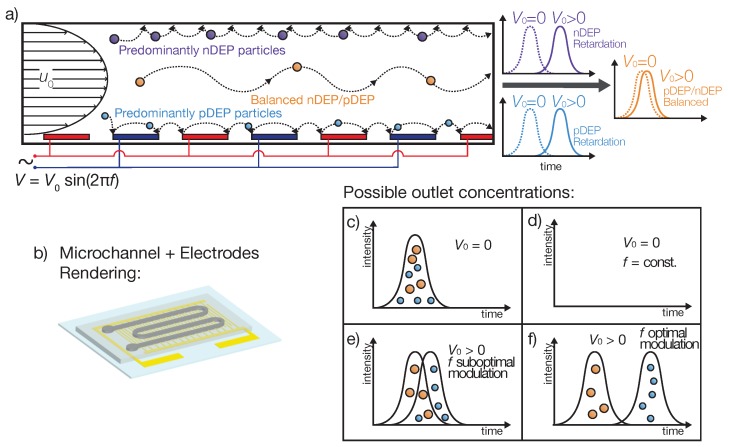
(**a**) Sketch of the DPC separation experiments. (**b**) Sketch of the DPC separation column. Meandering PDMS microchannel sealed by interdigitated electrodes on a glass chip. (**c**–**f**) Different possible outlet concentrations for DPC. (**c**) Without voltage, no retardation of the particles occurs, and both fractions elute at the same time. (**d**) When a voltage is applied and the frequency is fixed, the particles are trapped in the column due to DEP and will not exit the channel. If the frequency is modulated, a chromatographic separation occurs (**e**), which can be optimized by changing the frequencies and voltage (**f**).

**Figure 2 micromachines-11-00038-f002:**
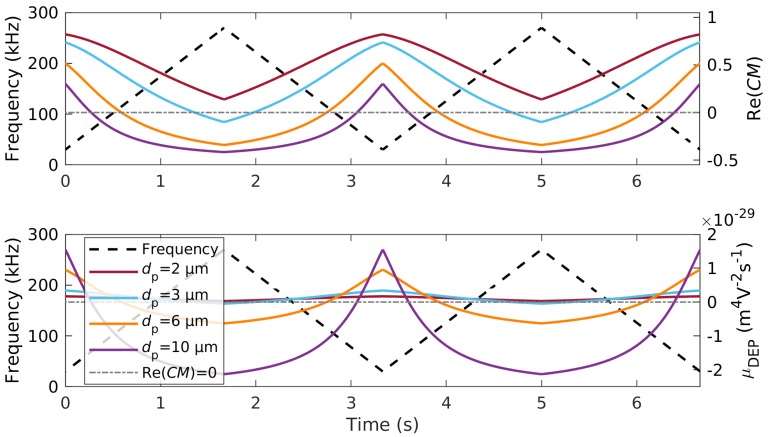
Real part of the Clausius–Mossotti factor Re(*CM*) (top) and dielectrophoretic mobility $\mu_{\mathrm{DEP}}$ (bottom) of four different polystyrene particles over time for two full cycles (right ordinate axis of diagram). The modulated frequency is shown as well (left ordinate axis). Particles suspended in DI water with $\sigma_{m}=$
1.2
μS
cm^−1^, $K_{S}=1\;nS$, and $\varepsilon_{m}=78.5$, calculated with Equations (3) and (4).

**Figure 3 micromachines-11-00038-f003:**
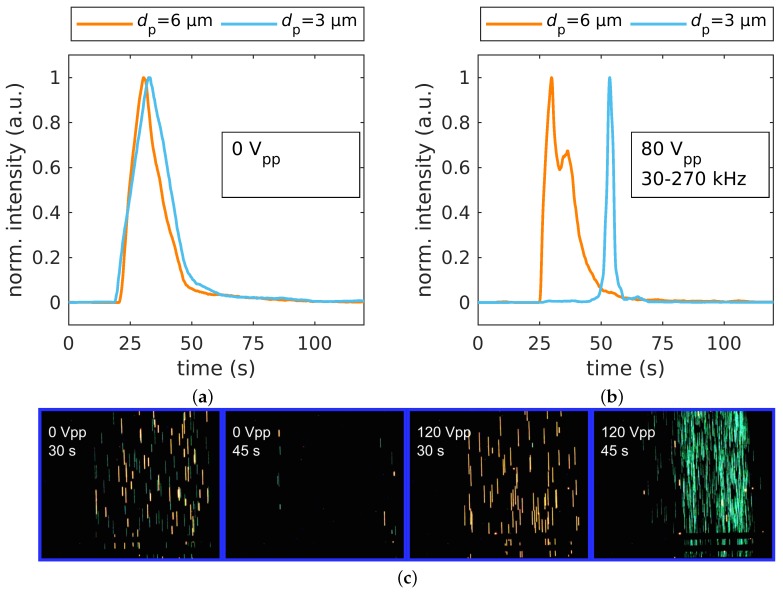
(**a**) Separation of 3 and 6 μm polystyrene particles: fluorescence intensity over time for no applied voltage, $R_{s}=0.17\pm 0.06$ (N=4), and (**b**) when applying 80 V_pp_ at 30 kHz–270 kHz with a modulation frequency of 300 mHz, $R_{s}=3.60\pm 0.31$ (N=4). (**c**) Single frames of different times of 3 μm (yellow-green) and 6 μm (orange/red) fluorescent polystyrene particles (brightness and contrast are adjusted for better visibility).

**Figure 4 micromachines-11-00038-f004:**
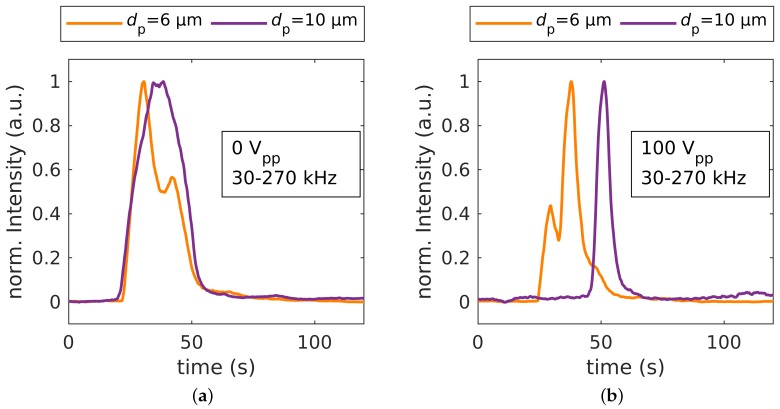
(**a**) Separation of 6 and 10 μm particles: fluorescence intensity over time of without applied voltage, $R_{s}=0.21\pm 0.19$ (N=4), and (**b**) with application of 80 V_pp_ at 30 kHz–270 kHz with a modulation frequency of 300 mHz, $R_{s}=1.95\pm 0.33$ (N=4).

**Figure 5 micromachines-11-00038-f005:**
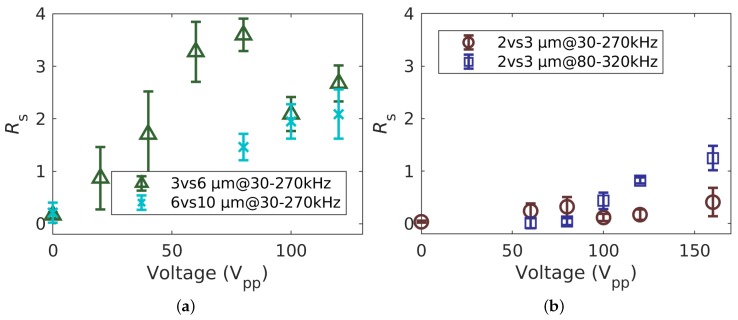
Resolution $R_{s}$ of DPC over applied voltage for different particle suspensions and frequencies. (**a**) 3 μm vs. 6 μm and 6 μm vs. 10 μm PS particles at 30 kHz to 270 kHz. (**b**) 2 μm vs. 3 μm PS particles at 30 kHz to 270 kHz and 80 kHz to 320 kHz.

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
