# Peer review of "Polarizability-Dependent Sorting of Microparticles Using Continuous-Flow Dielectrophoretic Chromatography with a Frequency Modulation Method"

_micromachines, 2019, doi:10.3390/mi11010038_

Round 1

Reviewer 1 Report

This manuscript presents a study of a chromatographic-like separation of polystyrene particles driven by dielectrophoresis. Overall, the manuscript is difficult to follow, the experimental evidence seems to be scarce, and the novelty of the approach is not clear. Many conclusions are not adequately supported by experimental evidence, nor by modeling work, which makes it difficult to assess what is a confirmed conclusion and what is speculation.

For instance, in line 175, the authors conclude that a balance between nDEP and pDEP is the reason to observe roughly the same intensity distribution of the 6 um particles with and without an electric field. This is a strong assumption, and there seems to be not much evidence in the manuscript that supports this claim. Can it be that the band of the 3 um particles is thinner because they are attracted/trapped by pDEP close to the electrodes? Can large particles be eluted first because they exhibit pDEP for less time? These hypotheses may also explain the observed behavior, but is hard to draw a conclusion without further experimentation. This work can be largely benefited by conducting more experiments, or by an extensive modeling work.

Some other contradictory statements or conclusions not supported by evidence are shown next (this is not a comprehensive list). In line 178: “at higher voltages, particles move far enough for them to hit either the electrode array, or channel ceiling, causing trapping and considerable retardation.” Was this trapping observed? How?

The same is true for (line 195) “we assume that, due to the higher frequencies, an increased negative dielectrophoretic movement pushes the particles away from the electrodes and thus towards higher fluid velocities.” Was this confirmed by any means?

Interestingly, the authors discuss the forces acting on the particles and the resulting velocities. However, they center they discussion on the component orthogonal to the flow. The parallel component is never mentioned nor discussed, but it should have a contribution on particle movement. This reviewer believes this work can be largely enhanced by extensive modeling work.

Below are some other comments that need to be addressed before the manuscript is accepted for publication.

In line 161, the authors describe the separation of 3 and 6 um particles for several electric potentials. They only show the results at 80 V (Fig. 3), and refer the reader to the supporting document to see the complete dataset. However, this information is missing. The supporting document only has information on the separation of 2 and 3 um particles. In this case, it is difficult to confirm the observations described by the authors.

In line 71, the authors discuss the regions with low (edges of electrodes) and high (top of the channel) fields, and mention that particles can be either trapped at the edges of the electrodes (by pDEP) or at the top of the channel (by nDEP). While particles can be trapped (or slowed down) by the “attractive” force towards the electrodes by pDEP, what is the mechanism for trapping them using the “repulsive” force of nDEP? Most studies discuss a deflection of the particles, not trapping per se. This is confusing.

In line 77, the authors mention that particle-wall interactions can be strong enough to observe particle adhesion, and that the DEP force should revert direction for these particles to be released. Considering the large size of these particles, have the authors observed any adhesion to the top wall (ceiling)? This seems to be surprising. If not, why was this mentioned?

As far as this reviewer understands, the metallic electrodes are isolated from the channel by a thin PDMS layer, and this is why the authors can apply such large electric potentials without observing electrolysis. However, the thickness of this layer is a random variable, not easily controlled, and it should have a large effect on the induced electric field inside the channel. Was the current experimentally measured? How many chips were tested? What was the variation between different chips? Were results reproducible?

An important limitation of the proposed approach is that large particles have a cross-over frequency only at very low conductivities, so that the separation of particles with a similar size cannot be easily resolved. Is there a reason why the authors did not consider, for example, separating a 100 nm particle from a 500 nm particle? Why to stick with the large particles, even if the separation seems to be poor?

In line 105, the authors mention that larger particles should be eluted earlier because the carrier flow has more time to transport these particles a greater distance per cycle. This description is not clear. Is this related to a retention by pDEP?

The authors speculate that their system can be used for the separation of 2, 3 and 6 um particles, while the manuscript presents not enough experimental information to support this claim. It may be the case that it is possible, but it may also be the case that is not. This should be removed from the manuscript, or more experiments/modeling should be provided.

Although the operation of the chip is described to a good extent, the chip operation is never schematically shown. The schematic can be seen in Figure 2, but is never referred in the text. Adding a schematic figure of the chip and the movement of particles under pDEP/nDEP could help the readability and understanding of the reader.

In Fig. 1, what are the units of the DEP mobility? Why are the CM factor and DEP mobility of 10 um particles plotted if they were not used?

In Fig. 2, what happens when the frequency is modulated? This should be represented in (d), and the figure’s label mentions that there is a separation, but the figure is blank.

In Fig. 4 from the supporting document, which one is the 3 um particles?

The introduction seems to be limited, since many studies closely related to the proposed approach have been omitted, and several ideas are disconnected from each other. There is not a natural flow in the introduction.

A deep proof-reading is recommended, since there exist several typos are present (e.g., “graident" instead of gradient) and a few phrases/ideas are incomplete through the manuscript.

Reviewer 2 Report

The manuscript focuses on a new approach for DPC based on differences in the dielectrophoretic mobilities and the crossover frequencies of polystyrene particles. It is in a good shape and the conclusions and results are supported by the analysis presented but also suffer some problems. Therefore, this manuscript can be accepted for publication after some minor revisions as suggested below.

In introduction part of the manuscript, more detailed info in literature review is required. Authors reviewed some methods and then just directly concluded that all these methods suffered problems such as being limited in selectivity and lacking the applicability for a wide range of particle mixtures etc. More details or evidence is necessary to testify authors’ judgement on others’ work. Maybe include a table to tabulate specifications of others’ work.

The novelty of this work should be addressed. Authors need to include and talked more about the why their work stands out. For examples, authors can discuss the features or advantages that their work has over others’ work.

Legend of Figure 1 needs to be relocated. It blocked the second half of Figure 1. Also, Figure 1 illustrates the changes of CM factor (real part) and DEP velocity with various time and frequency. It seems that authors are using a triangle wave as carrier wave. Please specify the voltage and frequency range of the applied signal in both text and figure. At last, I would recommend drawing 3D plots for both plots in Figure 1 to remove misleading info, because both CM factor and DEP velocity have two independent variables, time and frequency.

Reviewer 3 Report

The authors in the present article illustrate a novel technique for binary particle separation using dielectrophoretic particle chromatography (DPC). Electric field frequency modulation is utilized to generate positive and negative DEP to attain particle sorting. The manuscript is well written with adequately organized sections, clear figures formatting, and meaningful captions. Minor English language changes needed.

Author Response

We thank the reviewer for his nice comment. We conducted multiple additional loops of proof reading and rewrote some paragraphs to increase the readability and reduce grammar/spelling mistakes.

Reviewer 4 Report

Comments for micromachines-650056
It is proposed in this manuscript a modification of the so-called dielectrophoretic
chromatography for the sorting of microparticles via frequency modulation. The idea
was demonstrated by sorting polystyrene particles of two different sizes with different
diameters (2 μm versus 3 μm, and 3 μm versus 6 μm). Although such an idea is not
novel for researchers working in dielectrophoresis, it could possibly provide an
alternative for dielectrophoretic chromatography application. I think that the
manuscript could be published provided the following items are addressed properly.
(1) The authors performed their experiments in a micro channel with width 2 mm,
height 80 μm and length 17 cm, at a volume flow rate 5 mL/hr, which implies that
the average axial speed along the channel is 0.00868 m/s, and it takes about 1.96 s
on average for a particle to flow through the device. Such a time, 1.96 s, is less than
the time scale, π (= 3.1416 s), associated with the frequency modulation in Figure
1. This implies that the authors were performing just “partial” frequency modulation.
It is suggested that the authors should perform further experiments with some
smaller volume flow rates or shorter period of frequency modulation. I think the
ratio of the above time scales could be an interesting parameter for optimizing their
proposed method.
(2) The applied voltages in the experiments are rather high, of order 100 V (up to 160
V), and the electric field magnitude is of order 106 V/m for an electrode width 100
μm, in the manuscript. I am wondering whether electrolysis might occur? Also, the
electro-thermal effect could be significant when the application was performed in a
medium at some higher conductivities, which could happen for the application of
sorting bio-particles.
(3) It is interest to see the application of the present proposed method could be
demonstrated through the sorting of different types of particles, say, different cells.

Round 2

Reviewer 1 Report

The authors have successfully addressed all my previous comments. The revised version includes a restructuration of the whole manuscript, and new experimental evidence and modelling work to support the claims. The updated manuscript is now clear, and easier to follow. Figure 1 (schematic operation of the device) was much needed. I have no further comments at this time.

Reviewer 4 Report

The authors have responsed my comments properly, I do not have any further comments.